# Caregiver Reports of Screen Time Use of Children with Autism Spectrum Disorder: A Qualitative Study

**DOI:** 10.3390/bs9050056

**Published:** 2019-05-22

**Authors:** Anja Stiller, Jan Weber, Finja Strube, Thomas Mößle

**Affiliations:** 1Criminological Research Institute of Lower Saxony (KFN), 30179 Hanover, Germany; Jan.Weber@kfn.de; 2MediClin Deister Weser Clinics, 31848 Bad Münder, Germany; Finja.Strube@mediclin.de; 3State Police College of Baden-Wuerttemberg, 78054 Villingen-Schwenningen, Germany; ThomasMoessle@hfpol-bw.de

**Keywords:** media use, children, adolescents, autism spectrum disorder (ASD), leisure time

## Abstract

Screen based media have progressively become an integral part in the daily lives of children and youths with and without autism spectrum disorder (ASD). However, research that exclusively pursues the functionality of screen media use of children with ASD is extremely rare. Through a triangulated approach, the present study aims to fill this gap. We conducted 13 interviews with parents of children with ASD and supplemented this interview–study with an online survey including parents of children with ASD (n = 327). Children with ASD mostly used screen media (especially television) for their wellbeing, which is associated with chances and risks. Based on the parental interviews it is suggested that the media usage of children with ASD should be supervised. The results are discussed in terms of their practical implementation.

## 1. Introduction

### 1.1. Screen Media Use during Childhood

Screen based media have progressively become an integral part in the daily lives of children and youths with and without autism spectrum disorder (ASD). A current nationally representative parent survey by [1] showed that in the United States more than half of the 6- to 17-year old children with or without ASD use screen media more than two hours per day. With reference to TV consumption, most studies show that children and youths with ASD spend a maximum of two hours per day watching television [2,3], including more frequent use on the weekend [2]. Regarding the playing of video games, some studies show frequent media usage behaviour in children and youths with ASD [4,5,6], whilst other studies found that computer usage was a generally less frequent activity [7,8].

In general, (problematic) screen media use during childhood can be linked to a number of negative outcomes in child development, for instance poor academic performance [9,10], adiposity [11,12,13], low sleep quality [14,15,16] or attention problems [17,18,19]. In order to differentiate between problematic and unproblematic screen-based media use, Bleckmann and Mößle postulated that the three dimensions, time, content, and functionality, must be considered [20]. That said, problematic screen-based media use always entails an above average usage pattern regarding all named dimensions [20].

### 1.2. Three Dimension of Screen-Based Media Use: Time, Content and Function

For the time dimension, many health authorities, practitioners, and experts recommend that children under the age of two or three should not use screen media at all [21,22,23]. Referring to children aged two years and older a maximum consumption of two hours per day is suggested [21]. However, according to a recent survey by German researchers, a maximum screen media consumption of 30 min per day for children attending kindergarten, and a maximum of one hour per day for 7 to 12-year-old consumers is suggested [24]. At the time of writing, specific recommendations for children and youths with ASD were not available.

With regard to the content dimension, several institutions committed to the assessment and age rating of films, videos, and computer games can be identified across Europe (e.g., Germany: Freiwillige Selbstkontrolle der Filmwirtschaft GmbH (FSK), Unterhaltungssoftware Selbstkontrolle (USK); England: British Board of Film Classification (bbfc); Cross-European: Pan European Game Information (PEGI)). Disregarding their differing approaches, they collectively aim to protect children against unsuitable or harmful content. Age ratings have the potential to support consumers and particularly parents in their purchase decisions. Focusing on selected computer game genres, for instance online multiplayer video games, specific characteristics have been found to encourage an excessive gaming behaviour (e.g., social integration, intermittent reinforcement, [25]). However, in their cross-sectional parent survey, the authors observed that the majority of youths with ASD are more likely to never engage in playing online multiplayer video games in contrast to their typically developed control group [2]. Nonetheless, certain characteristics of conventional video games also make them appealing to young people with ASD: the absence of social interaction, consistency and predictability, active control and challenges, the opportunity to escape into a fantasy-world. For some children with ASD, video games become an obsessive interest [26,27].

Relating to functionality as the third dimension of screen-based media use, it was observed that media are utilized by children and their parents for different reasons. Among others this includes parents of children without ASD using screen media, especially television, as a babysitter [28,29,30] or as reward or punishment [31,32]. Furthermore, it is occasionally observed that children without ASD use screen media in order to relax. They might also turn to screen media when they are bored and seeking entertainment; others might use screen media as a background noise, for educational purposes, for homework assignments or while spending time with friends or family [28,30,32,33]. For the functionality dimension in children and youth with ASD, some interesting findings concerning the social function need to be highlighted: Available evidence suggests that children with ASD use video games as an opportunity to interact with peers and thereby feel accepted [26,34]. In contrast, some children with ASD use video games and other screen media content in order to avoid social interaction or to compensate for a lack of friends [26,35]. Other studies with and without children with ASD indicate that screen media use is a preferred joint activity within the family [36,37,38] and also when spending time with friends [39]. Of further interest are those findings that show positive effects of media use in children with ASD. Studies have reported, for instance, that the exposure to electronic media could cause positive behavioral changes in children with ASD [38] or have a positive influence on educational skills [21]. Furthermore, video games especially have a beneficial impact on social [8,26,34] or communicative [26] skills in children with ASD.

### 1.3. Objectives

To the best of our knowledge, the currently available research only marginally addresses the relevance and functionality of screen-based media use of children with ASD. Research that exclusively pursues this issue is extremely rare. Furthermore, a review by [40] emphasizes that more research is needed in this field. The present study aims to fill this gap by further exploring the functionality of screen media use in the lives of children and youths with ASD. A better understanding of this field is crucial for the development of specific preventive approaches or group activities for children and youths with ASD as well as for positive uses. Therefore, in a second step, we draw practical implications based on our analysis of the functionality of media use for children and youths with ASD.

## 2. Methods

For the present study ethical approval has been obtained through the ethics committee of the University of Hildesheim, Department 1: Education and Social Science. All subjects gave their informed consent for inclusion before they participated in the study.

### 2.1. Recruitment and Procedere

#### 2.1.1. Qualitative Survey

The study is based on a triangulation of data from qualitative and quantitative sources. First, thirteen parents of children and youths with ASD were interviewed between October 2015 and January 2016 using a semi-structured interview. The interviewees were recruited through public announcement and flyer distribution at an Autism Center in Northern Germany. This facility provides kindergarten and schooling opportunities for children and youth diagnosed with ASD. In Germany people with ASD are entitled to benefit from services provided by the public health and social welfare systems. This also implies specific facilities (e.g., schools, workshops). With reference to the German legislation for the disabled, a person with ASD also has the opportunity to be assessed for a disability grade, where a disability grade of 50% or higher is considered severely disabled (for further details see [41]). Other parents participated in the interview due to word-of-mouth recommendation as other attending parents shared their experience within the Autism Center.

The semi-structured parental interview focused on the domestic and off-site leisure-time behaviour of children with ASD, media consumption of these children as well as relevant conditions (e.g., content, rules, and behaviour during media use). In addition, intra-family conflicts and other problems with respect to the children’s media use were addressed (e.g., impact, realisation of rules). Since no differentiations between media types were made by the interviewer, conversations regarding media use encompassed all types of screen-based media. The interviews were conducted in German, audiotaped, and subsequently transliterated using the F4 software-package. The interviews lasted on average 122 min and ranged from 70 to 228 min.

At the end of each interview, interviewees were asked to complete a short questionnaire. Assisted by the interviewer, parents’ personal and sociographic information (e.g., gender, age, family status, current work situation), information about their child with ASD (e.g., diagnosis, language ability), and information related to the screen-based media equipment available for their child with ASD at home was assessed. Finally, the interviewees signed a declaration of consent.

#### 2.1.2. Quantitative Survey

Second, a parent online survey regarding the leisure time of their child with ASD was conducted between June and December 2017. In this survey, information on different media types was collected, which enhanced the qualitative survey and enabled a more differentiated picture. One open-ended question addressed the function of using the different media types (“Why do you think your child is watching TV or DVD/plays computer or video games/engages with the smartphone/is using the internet in the situations you specified in the question before?”). The participants had the opportunity to give a maximum of three different reasons for their media usage.

In order to recruit parents of children and youths with ASD, regional associations and autism therapy centres in Germany were contacted requesting their support for this project. The supporting institutions forwarded the link for the online survey, along with information about the project to parents and/or posted the link with related information on their homepage. Information about the project, including the corresponding link, was also distributed through flyers at the German Autism Congress in June 2017. Additionally, the link for the online survey was made available on the homepage of Autism Germany e.V., including details about the project. The survey was directed at parents of children or youths who have been diagnosed with ASD aged 17 or younger.

The online survey was conducted with the online platform SoSci Survey. Prior to participation, participants were informed about data security and anonymity, their participation by choice, and the possibility to stop filling-in the questionnaire at any time. The online survey did not start until the participants agreed to participate in the survey (“Yes, I would like to participate.”). A total of 327 parents took part in the survey.

### 2.2. Analysis

#### 2.2.1. Qualitative Survey

Analyses were conducted by means of qualitative content analysis, a method to analyse qualitative data in a systematic way [42]. Following a deductive paradigm and provided with coding guidelines, categorization was performed by two independent coders using MAXQDA (version 11). Each interview was analysed sequentially by each coder. In order to get assigned to a category, a single segment had to comprise at least one sentence up to a maximum of one paragraph. Some statements have been assigned to several categories. Cohen’s Kappa was determined with MAXQDA and SPSS (version 19, for generating of the subcodes see below) and used to denote consistency with a rating of >0.6 being considered substantial [43]. In this study, the categories “function” (coding guidelines: why media is used, purpose), and “shared screen media use” (coding guidelines: shared media use, e.g., playing computer games, watching a movie, cinema; involvement in media use) were of interest. The interrater agreement ranged from fair (κ = 0.401) to moderate (κ = 0.440) prior to correcting discrepant items. In this context it must be mentioned that MAXQDA included the length of the coded section when measuring the interrater agreement. Every unmatched subcode was discussed among the independent coders until 100% agreement was achieved. The opinions of both coders were treated equally.

Throughout qualitative content analysis, data was further summarized by paraphrasing, generalizing, and reducing separate paragraphs. Subcodes were developed inductively, as the paragraphs were analysed by one rater. Similar paraphrases were compressed, integrated, and compiled into subcodes. To ensure coherence, an independent rater coded all paragraphs using the given subcodes and coding guidelines (see Table 1). The levels of interrater agreement were substantial for each category prior to correcting discrepant items (κ = 0.622 up to κ = 0.766). Every unmatched subcode was discussed among the independent coders until 100% agreement was achieved. The opinions of both coders were treated equally.

#### 2.2.2. Quantitative Survey

The answers of the participants were grouped by two independent coders using deductive codes and coding guidelines generated by the qualitative analysis (see Table 1, code “function”). Cohen’s Kappa was determined with SPSS (version 19) and, in general, was substantial prior to correcting discrepant items (κ = 0.744). Even if the different media types are considered in detail, the levels of interrater agreement were substantial for each type (TV/DVD: κ = 0.712; computer or video games: κ = 0.683; smartphone: κ = 0.772; internet: κ = 0.783). Any unmatched free text information was discussed among researchers until 100% agreement was achieve. The opinions of both coders were treated equally.

## 3. Results

### 3.1. Qualitative Survey

The crucial characteristics of the interviewees and their respective child are summarized in Table 2. The chart also provides an overview on the screen equipment in the home of the participants as well as in the bedroom of the child.

#### Main Results

Results are presented along the main categories “function” and “shared screen media use”. The category “shared screen media use” was considered as it takes up individual aspects of a subcode in the category “function” and therefore provides a more differentiated picture. Some statements might relate to several perspectives on a phenomenon. Sometimes a comment can be considered from different perspectives. Thus, a single comment might appear in different subcodes. With regard to the main categories, 18 subcodes were identified (see Table 3). Table 3 include also some parental statements exemplary for each code.

Function: A total of 166 comments related to the category “function” have been identified. The majority of comments in the interviews referred to the fact that screen media was used for ‘wellbeing’ purposes by the children (n = 43). Other functional meanings were attributed to the ‘advancement of other functional areas’ like language or creativity (n = 16) or ‘social communication and exchange’ (n = 16). For the other nine subcodes please refer to Table 3.

As outlined above, screen media usage implies a wide range of functionality for children affected by ASD and their families. However, beyond the described functional meaning, screen media usage also seems to play an important role with regard to shared family time. The following category refers to its qualitative significance.

Shared screen media usage: In total, 78 statements were assigned to this category. Out of six, the rather unspecific ‘shared family time’ turned out to be the most important subcode, as a total of 40 paragraphs were found to relate to it (n = 40). Some comments reflecting a ‘(mutual) topic of conversation’ triggered by shared screen time (n = 9). Shared screen media use also plays an important role in the context of ‘evening rituals’ (n = 9). For the other three subcodes please refer to Table 3. Interviewees with more than one child were usually assigned to this category.

Regarding the possibility of age-related differences within the context of media usage, the data revealed that adolescents used media predominantly for social communication or to raise their social acceptance. In contrast, younger children’s media usage occurred more frequently in relation to, what one could call a babysitter function, or as an evening ritual. But some subcodes were also mentioned independent of children’s age (esp., wellbeing, learning tool, safety, monitoring).

### 3.2. Quantitative Survey

The crucial characteristics of the participants and their respective child are summarized in Table 4. The chart also provides an overview on the screen equipment in the home of the participants as well as in the bedroom of the child.

#### Main Results

The findings from the qualitative analysis above are reflected in our analysis of the free text information of the online survey. All 12 subcodes of the code “function” generated by the qualitative analysis were identified in the free text information (see Table 5). As with the qualitative analysis, participants most often answered “for wellbeing” to the question: why is child their using screen media (63.5%)? However, using screen media as a “social learning tool”, as an “advancement of other functional areas”, as “background noise” or as a “babysitter” was mentioned more often in the interviews. In contrast, the subcodes “source of information” and “joint family activities” play a more important role for the participants of the online survey. The consideration of the diverse media types provides a more differentiated picture: In order to relax children predominantly used playing computer or video games (75.8%) followed by watching TV/DVD (64.0%), using the smartphone (57.9%) or using the internet (53.6%). The internet was more often referred to as a source of information (32.9%), followed by other screen media devices (smartphone: 13.5%, TV/DVD: 9.1%, PC/video games: 3.1%). Furthermore, spending time with the family was more relevant in the context of watching TV/DVD (12.5%) compared to the use of other screen media (PC/video games: 3.1%, smartphone: 1.2%, internet: 0.5%). The smartphone served as a security device (11.7%), whereas PC or video games (4.9%), the internet (3.4%) or the TV (3.1%) were used less frequently for this purpose. Social communication/exchange is also a rather functional component of the smartphone (10.5%), much less so regarding the use of PC or video games (3.1%), the internet (2.9%) or the TV (0.3%). Furthermore, TV/DVD (5.7%) was more often used as a “babysitter” PC/video games (1.3%), smartphones (0.6%) or internet (0.5%).

## 4. Discussion

### 4.1. Exploring the Functionality of Screen Media in the Lives of Children and Youths with ASD

In general, screen media was mostly used for wellbeing, with playing computer or video games being used particularly often for this purpose among children with ASD. Almost all interviewees mentioned this topic and it was by far the most frequent statement within the online survey. Screen media usage for wellbeing purposes among children without ASD has been reported before [32,44,45]. However, especially for children with ASD this wellbeing is of extensive significance: Regarding the sensory overload in pupils with ASD [46,47,48,49,50], it can be speculated that focusing on one stimulus (screen) might help fading out other stimuli for a certain time, and thus relaxes other senses. Several studies have shown a preference for visual stimuli, especially via media, in children with autism [8,51,52]. Other studies reported that while playing video games [34] or watching TV [8] children with ASD fade out sensory challenges or environmental distractors.

Based on these findings, it may be suggested that screen media might have the potential to positively influence children with ASD through rebalancing their personal overstrain caused by the great amount of environmental stimuli they encounter. This might be of further relevance considering that four children in the interview study used screen media as a background noise, for example as a sleeping aid. As highlighted by other studies, the background noise of screen media is also commonly used among children without ASD [30]. It can be suggested that permanent background noise (e.g., white noise through television) may particularly help children with ASD to fade out other stimuli as mentioned above—in the auditory instead of the visual sense. Future research is needed regarding the wellbeing effect of screen-based media in children and youths with ASD.

The present study demonstrates that screen media are used to support the development of certain skills (e.g., language, social competence). Seven interviewed parents reported that their children are imitating voices or noises of the screen media as well as watching social stories from which they can learn. In the online survey this point seems to be less crucial. The functional “learning aspect” in using screen media might become more obvious while having a free conversation about one’s own child (interview) compared to a situation in which one is asked to complete a questionnaire (online survey). However, some studies examined the development or optimization of supportive strategies which use media and particularly videos as a model learning component. Ref. [53] described different computerized programs, which focus on social communication skills (e.g., language, emotion recognition) and highlight the positive effects computer technology can have in this context. In Interview 7 of this study, for example, the child benefited from bedtime stories. Bedtime stories are usually characterized by their simplification of otherwise complicated social interactions, problems or other phenomena. It can be suggested that the reduction of complexity helps children to understand the point of the story and to learn from the presented role-model and its behaviour. Due to their obvious messages, bedtime stories are easy to understand and may work even better for children with ASD. In this context, other studies have reported the positive influence that electronic media can have on social or communicative skills [8,21,26]

Another important finding of the present study is that almost all interviewees mentioned that screen media time is often perceived as family time (it was also one of the most frequent statements in the online-survey), for instance when the whole family is attending television events together, enjoying television evenings or when screen media content becomes a shared topic of conversation (e.g., at dinner). This is also reported in other studies on children with [36] and without ASD [28]. The participants in the present online survey stated that watching TV/DVD is especially used as shared time with the family. Spending quality time together with a child with ASD is not always easy to achieve. Studies indicate that parents of children with ASD feel more stressed than parents of children without ASD [54] or compared to parents of children affected by chronic illness (e.g., cancer, diabetes), or at-risk (e.g., low income, subclinical behavioral problems) for behavioral and/or mood disorders [55]. Thus, spending time together in a relaxed (screen) atmosphere may offer parents an opportunity to brighten everyday life and increase the quality of family life. This becomes particularly relevant when taking into account that media can offer a common topic of conversation. However, almost half of the interviewees mentioned that it is important to attend to the child during screen media use in order to answer any questions from the child (educational support) or to control the media consumption of the child (monitoring). This may be of particular relevance for children affected by ASD. Drawing for example on the report of Interviewee 7 or 13 highlights the need of an accompanying person to help put things into perspective. Thus, it can be important for children with ASD to have the opportunity to address questions that might come up during their media consumption.

This point is even more important considering that children with special needs are more likely to become victims of cyberbullying than children without special needs [56,57,58]. In this context, [59] found that parental mediation had a buffering effect on the relationship between cyber victimization and depression in children with ASD. It may be favourable for parents to be aware of the content of their child’s media use as well as related problems in order to be able to consider it in the assessment of their child’s behaviour, and thus react to it in an adequate way.

In the present study, screen media was also reported as being used for communication and social exchange, for instance while doing homework, by four interviewees and some participants of the online survey. Particularly smartphones are used for this purpose. In the interviews, parents appeared pleased by their child’s interaction with other children via chats or social apps, even though this was mostly limited to communication in a restricted scope and rarely lead to face-to-face-encounters with friends. However, messaging features seem to play a vital role for children without ASD [60,61,62,63], but possible also for children with ASD. In this context, some parents highlight what they consider a positive side effect. According to them, the respective communication that comes with it helps their children to overcome social isolation.

Other children of the present study are using screen media in order to improve their sense of safeness, on the one hand as a means to contact others and for communication, e.g., by calling their parents. The participants of the online survey ascribed particular importance to smartphones as being a safety device; also because they provide the opportunity to make emergency calls. One interviewed mother of the present study mentioned that her son would not independently approach a stranger to call home, but he would call home with his own phone.

In the present study, screen media was also used as a source of information by some children described in the interviews and in the online survey (the second most frequent statement), e.g., watching TV to get factual information or using the internet to look for information. In times of proceeding digitalization this is what one would expect, especially since available evidence suggests that children without ASD use screen media for educational purposes or homework assignments [32,33].

Furthermore, the present study indicates that some parents at times use screen media to gain some free time, or to take care of the needs of their other children. This “babysitter” function of screen media has also been observed in studies with parents of children without ASD [28]. However, screen-based media becomes problematic if parents use screen-based media as a digital babysitter [20]. This becomes particularly relevant when the above-mentioned suggested need for educational support and monitoring during screen use is taken into account. It was also observed in the present study that screen media were occasionally applied as an instrument to reinforce desired behavior by some interviewees. According to studies with parents of children without ASD, this seems to be common as results show that screen media, especially television, are used as a reward or punishment tool [31]. However, using screen-based media as a reward, or its withdrawal, as a sanction could entail a problematic screen-based media use [20], whereas the other dimensions (time, content) must be considered as well.

Some parents of children of the present study reported that screen media helps them to gain a certain status, especially to become part of a group (social acceptance). Interviewee 10, for instance, expressed her belief that providing high-end devices to her child will help him maintain his social inclusion. It therefore seems possible that screen media devices support children with ASD not only in interacting with peers [26], but also in retaining the social status within their peer group, which may be associated with children’s desire for affiliation. For example, in their study [64] observed that children with ASD generally want to have friendships. However, older children rather avoid social interaction due to their negative experiences [64]. In their experimental study, [28] concluded that children with ASD do have a wish for social interaction, though it might not be at an explicit (measured by the Social Interaction Scale) but instead at a rather implicit (measured by the Face Turn AAT) level. That is, although there is a (implicit) desire for social interaction, interacting in a direct way might be too complex for many of children with ASD [28]. In general, the desire for affiliation is a natural part of children’s development and a core source of motivation in humans of all ages [65,66,67,68]. Even if children with ASD are less socially motivated [50,69,70,71], there is evidence that some children with ASD feel lonely [72,73], and are objectively more excluded than children without ASD [74]. Furthermore, if their problems in social interaction [75,76,77,78,79] are taken into account, affiliation is more difficult to achieve for children with ASD. Thus, media use might potentially provide easier access to social experiences and it may consequently increase the sense of safeness in social interactions. In the cross-sectional study the authors observed a positive association between social media use and quality in friendships in adolescents with ASD [80]. Contrary to the desire for social affiliation, some studies found that children with ASD use screen media in order to avoid social interaction or to compensate for a lack of friends [26,27,35]. This was not be confirmed in the current study. Though one needs to consider that parents had been asked directly to report about their child’s functionality of screen media. It seems likely that the compensation for a lack of friends therefore is not among the first functions that comes to parents’ minds.

For some children in the present study, screen media plays a major role in daily life. This also bears the risk that the child is preoccupied with thoughts about using screen media, which might result in problems (e.g., excessive use, addiction). In this context it is important to consider the risks and chances involved in screen media use including the additional dimensions time and content. That is, the trade-off between risks and chances might not always be easy to determine. Particularly online multiplayer video games encourage an excessive gaming behaviour due to specific computer game characteristics (e.g., social integration, intermittent reinforcement) [25]. However, the majority of youths with ASD are more likely to never engage in playing multiplayer video games in contrast to their typically developed control group [2]. Despite this and disregarding the suggested positive attitudes towards screen media as a tool to overcome social isolation as reported in the study, the risk of negative outcomes [12,14] must be taken into account.

### 4.2. Practical Implications, Limitations and Future Prospects

Screen media use involves risks and chances for children with and without ASD. According to the findings of the present study, it is suggested that children with ASD can benefit from screen media by using it for wellbeing purposes and for the improvement of language skills and other social competencies. Thus, it seems beneficial to include screen media in (selected) therapeutic approaches when working with children with ASD. Further research is needed in order to help identify best practice strategies.

The present study also suggested that screen media may help children with ASD to feel safe (e.g., calling) or socially accepted (fulfilling the desire for affiliation). The (resulting) feeling of safety may be seen as a chance for children with ASD to compensate a possible lack of confidence. However, it is important to note that screen media must not be the only strategy to achieve affiliation but that a child with ASD should also be strengthened in other areas; keeping in mind that studies have shown lower self-esteem [81] and low social competence [82] in persons without ASD who are addicted to video games. Furthermore, two longitudinal studies with youths without ASD observed that youths with higher levels of loneliness [83] and a lack of success experienced in real life [84] are particularly at risk of developing an excessive and problematic usage of video or computer games. However, monitoring and supporting their children in their screen media use regarding time, content and functionality seems to be more realistic when parents of children with ASD have time for themselves in other situations. If there are few alternatives to have time for themselves, parents of children with ASD use the time their child sits in front of the screen for themselves as well. This is understandable when considering that parents of children with ASD experience more stress than other parents [54,55].

Thus, regarding leisure activities, there is a need for alternatives to screen media use for children with ASD. [85] reviewed eight studies that addressed the level of quality of life in adults with ASD and associated factors. They summarized that the more participation in leisure time activities, the higher the reported quality of life in adults with ASD. To the best of our knowledge, there is no study that examined the relationship between leisure time activities and quality of life in children or youths with ASD. However, studies indicated that through watching television/movies other leisure activities (e.g., outdoor play, music) are neglected [2,8]. Future research is needed regarding the leisure time activities of children with ASD and their impact on the quality of life.

Any conclusions drawn from this study must be interpreted in the context of several methodological considerations. The present study is based on parental (mostly maternal) information about their child (external assessment); the children were not interviewed themselves. Though a participatory approach would be more appropriate, there is no tradition in Germany of involving (handicapped) children in research yet. Considering related ethical issues and methodological challenges, it was decided to collect parental information for this exploration first. However, parent surveys are a reliable source of information about the living environment of children, in particular children with special needs. There are a lot of studies that generate their data on children with ASD through parent surveys [1,86,87,88,89,90,91]. Nonetheless, future studies would benefit from involving the children’s perspective. Data might be collected through (participatory) observation, interviews or surveys in diary format. Furthermore, the recruitment of the participants was realized through public announcements, flyer distribution, and word-of-mouth recommendation (interview study) as well as through cooperating with regional associations, autism therapy centres, and through placing the link for the online survey on the homepage of Autism Germany e.V. (online survey). The majority of children reported on in the online survey were diagnosed with Asperger syndrome; infantile autism or atypical autism were underrepresented. In the interview study there were no children with atypical autism, but the other two diagnoses were in balance. Almost all of the children had verbal communication skills. Readers should be cautious when generalizing the results presented here to other populations of children with ASD. Finally it must be highlighted that despite the fact that it is generally well known that using patterns differ between age groups [5,92,93], the quantitative data in the present study does not facilitate a comparison of developmental trends. However, the qualitative data indicates that there are some differences between younger children and adolescent, but some subcodes were also mentioned independent of children’s age. Nonetheless, the number of interviews was limited to 13. Future research with a larger sample would benefit from analysing developmental patterns in order to generate a more differentiated picture (e.g., regarding the age or the severity of symptoms). In this context, a control group would be a good opportunity to compare the patterns of children with ASD with children without ASD.

## 5. Conclusions

The present study represents a first approach towards exploring the functionality that screen based media use has among children with ASD, and provides primary guidance to parents in dealing with their children’s media use. It is essential to note that regardless of the type of media, screen media seems to play an important role as an opportunity to relax. In this context, it appears necessary to supervise a child with ASD during media consumption, especially in order to answer questions the child may have as well as to include the media content in the understanding of the child’s behaviour. To meet this requirement, parents also need time for themselves. Therefore, it would be helpful to expand the offer of leisure activities for children with ASD in order to relieve parents in everyday life and thus reduce the parental stress-level. This provides parents with a useful opportunity to be qualitatively involved in their child’s media consumption and consequently in their daily life. Additionally, through an increase of the number of alternative leisure activities, children with ASD could potentially improve their quality of life.

## Figures and Tables

**Table 1 behavsci-09-00056-t001:** Coding guidelines (subcodes).

Code	Description
*Function*	
Wellbeing	Reduction of stress/tension, occupation, balance, fun, focus, calming down, timeout, vegging, everything to get lost in it
Social learning tool	Advancement of social skills (e.g., figures of speech, behavior with peers, dealing with conflicts)
Advancement of other areas	Advancement of other areas other than social skills (like language, emotion, creativity)
Social acceptance/be part of a group	Shared topics of conversation, status symbol, coolness, means of participation
Safety	In case of emergency (parents, child); control, revisions, no negative feedback, can be regulated by self, transparency/predictability, distraction, anxiety reduction
Purpose in life	High significance in life of the child, high meaning for child
Social communication/exchange	Communicate/discuss with other people, having social contacts and network, time with other people, agreements
Babysitter	Entertainment/occupation for the child, to have more time for self, managing chores, have some quiet time
Reinforcement	Media as an instrument, so that the child behaves in the desired way
Joint family activities	Shared time with the family, evening ritual
Source of information	Facts, information gathering, knowledge expansion
Background noise	Background or side noise, no total silence, several devices simultaneously
*Shared screen media usage*	
Shared family time	Use of screen media together with the family (e.g., watching TV, playing computer games)
Monitoring	Emphasizing, that the media use of the child is being monitored (e.g., use, content, time spent), it is taken care of nothing getting broken
(Mutual) topic of conversation	The media offer a topic of conversation, child exchanges ideas and thoughts with the family, child talks about it (one-sided, even if the other person doesn’t show interest in it), child involves family
Educational support	Child is supported, e.g., with internet research; explanation, e.g., to understand a movie (counter insurance)
Evening ritual	Shared screen media use as an evening ritual
Media use all alone	Engagement with screen media, preferably by oneself

**Table 2 behavsci-09-00056-t002:** Basic characteristic of the participants in the interview study and their child as well as the screen equipment (home, bedroom of the child).

Parents			Children			Screen Equipment		
**Age**	**Years**		**Age**	**Years**		**At Home**	**N = 13**	**%**
m	40.0		m	9.2		TV	13	100.0
range	26–54		range	4–17		Games console	5	38.5
**Gender**	**N = 13**	**%**	**Gender**	**N = 13**	**%**	Computer	1	92.3
female	11	84.6	female	1	7.7	Table	10	76.9
male	2	15.4	male	12	92.3	**Bedroom of the child**	**N = 13**	**%**
**Current work situation**	**N = 13**	**%**	**Diagnosis**	**N = 13**	**%**	no devices	**5**	38.5
working part-time or by the hour	7	53.9	Asperger Syndrome	6	46.2	TV	**2**	15.4
housewife/-husband	4	30.8	Infantile Autism	7	53.8	Games console	**1**	7.7
working full time	2	15.4	**Further diagnosis (e.g., ADHD)**	**N = 13**	**%**	Computer Tablet	**3** **3**	23.1 23.1
**marital status**	**N = 13**	**%**	no	9	69.2			
living with partner	12	92.3	yes	4	30.8			
living alone	1	7.7	**Current use of medication**	**N = 13**	**%**			
			no	9	69.2			
			yes	4	30.8			
			**Verbal communication**	**N = 13**	**%**			
			no	2	15.4			
			yes	11	84.6			
			**Siblings**	**N = 13**	**%**			
			no	2	15.4			
			yes	11	84.6			

**Table 3 behavsci-09-00056-t003:** Generated subcodes, their frequency of occurrence and some exemplary parental statements.

		n	Interview	Examples
*Function* *(n = 166)*	Wellbeing	43	1, 2, 4, 7, 8, 9, 10, 11, 12, 13	‘Like I said, very important for him, he can focus and calm down, gets lost in what he sees and he is occupied, so it’s sometimes very, very important for his emotional wellbeing.’ (Interview 12) ‘But sometimes it’s like she urgently needs it in order to get her stimulus response system back in balance, then she needs it too. If it [the content] fits, she is happy and balanced.’ (Interview 4)
	Advancement of other functional areas	16	1, 6, 9, 12	‘(…) he is picking up outstanding things from it. He gets all the numbers and figures and so on, he got all that from the iPad.’ (Interview 1)
	Social communication/exchange	16	9, 10, 11, 13	‘(…) communicating via WhatsApp, doing homework together, that’s what he does a lot, also, to compare with the whole class and help each other with certain homework.’ (Interview 13) ‘They play communicative parlor games on another platform. And considering that the result of that is, that he actually meets someone, I think, everything’s good.’ (interview 9)
	Social learning tool	12	7, 8, 10, 12	‘And then at some point, we started it [Das Sandmännchen] again, because there are pretty nice stories, also social stories. From which he can learn.’ (Interview 7)
	Safety	12	1, 2, 6, 10, 11, 12, 13	‘So, because he does not know, when he gets irritated, if he could go to the (school-) secretary in that moment and say “I want to call my mother”. That’s why he needs his mobile phone.’ (Interview 13)
	Babysitter	11	1, 2, 4, 7, 10, 12	‘(…) if I want to have some quietness and peace, he gets it, to be completely honest. If I don’t have the time to deal with him right now, because I am doing homework with the children, I am glad if he is busy with the iPad and I know he is not fooling around in that moment (…).’ (Interview 1)
	Joint family activities	11	2, 4, 7, 8, 9, 12	‘Sometimes it’s nice to watch some TV with her, just to spend time with her.’ (Interview 4)
	Reinforcement	10	2, 3, 6, 7, 9, 10, 11	‘From time to time as a reward for 10 or 15 min, then we try out movies.’ (Interview 6)
	Background noise	10	1, 4, 6, 8	‘If we turn it off [the TV] and so on, especially in his room, then anger (evolves), he would scream and so on, then, at night (the TV) must stay turned on all night long, otherwise he wouldn’t sleep.’ (Interview 6)
	Social Acceptance/be part of a group	10	9, 10, 11, 13	‘We thought about it a lot, that it is also a type of coolness and also when he goes to grammar school now, we are aware that he will get the newest and best devices from us very deliberately, may it be a smartphone or brands or whatever, to not let him get excluded because of those things too.’ (Interview 10)
	Purpose in life	8	9, 10	‘Yes, it is all about this [tablet] (…) sometimes, for instance, at school, when he doesn’t know where exactly it [tablet] is at home he gets stressed and would ask whether it is safe, or somehow he got really hysterical a few days ago because he thought, well I took it with me, so the school escort had to call me (…). In his head it absorbs about 90 percent, I think.’ (Interview 10)
	Source of information	7	8, 9, 13	‘Well, TV is actually his source of knowledge. Since he doesn’t read and also no non-fiction or specialized books, he uses it as an acoustic medium to get information.’ (Interview 8)
*Shared* *screen* *media* *usage* *(n = 78)*	shared family time	40	1, 2, 3, 4, 6, 8, 9, 10, 11, 12, 13	‘First he wanted to draw all the time, I just opened paint then, that’s this standard painting and drawing program. And then we drew trams. All day long we drew trams.’ (Interview 12) ‘Then I went to the Gamescom with Til and (…) to the Wars Identity Exhibition.’ (Interview 9)
	(Mutual) topic of conversation	9	7, 8, 9, 10, 11	‘We need to know somewhat what he does because he obviously talks about it a lot.’ (Interview 10).
	Evening ritual	9	3, 7, 9, 10, 12	‘So this Sandmännchen, this ritual, we do together. Dinner is finished by then and we all sit there [in front of the TV] and then we cuddle and smuggle and we watch it together.’ (Interview 7)
	Educational support	7	7, 11, 13	‘He also needs me to watch it as well as a backup (…) because he has a lot of questions on whether he understands things correctly or not.’ (Interview 13)
	Monitoring	7	6, 7, 9, 10	‘(…) also, it doesn’t make sense, he is either very hysterical and I don’t know why, because I didn’t see which scene I need to explain to him, or he doesn’t concentrate on it and does something else, then there’s no need to watch it at all.’ (Interview 7)
	Media use all alone	6	1, 4, 7, 8	‘Tuesday and Wednesday there is an afternoon of video-game playing. And that’s somewhat, because to be honest, I can’t stand it, because he sits in front of it playing [video games] all the time. So I am upstairs at my PC most of the times, handling some things I need to do.’ (Interview 8)

*Note*. The interviews were conducted in German. Despite our best efforts, interpretation-bias as a result of the translation to English language cannot be ruled out.

**Table 4 behavsci-09-00056-t004:** Basic characteristic of the participants in the online survey and their child as well as the screen equipment (home, bedroom of the child).

Parents			Children			Screen Equipment		
**Age**	**Years**		**Age**	**Years**		**At Home**	**N = 327**	**%**
m	42.2		m	10.94		TV (I)	163	49.8
range	26–70		range	3–17		TV	205	62.7
**Gender**	**N = 327**	**%**	**Gender**	**N = 327**	**%**	Games console (I)	124	37.9
female	290	88.7	female	58	17.7	Games console	97	29.7
male	27	11.3	male	269	82.3	DVD-player	251	76.8
**Current work situation**	**N = 318**	**%**	**Diagnosis**	**N = 327**	**%**	Computer (I)	309	94.5
working part-time or by the hour	156	49.1	Asperger Syndrome	167	51.1	Computer	32	9.8
housewife/-husband	60	18.9	Atypical Autism	111	33.9	Tablet (I)	239	73.1
working full time	55	17.3	Infantile Autism	49	15.0	Tablet	29	8.9
not working (e.g., student)	18	5.7	**Further diagnosis (e.g., ADHD)**	**N = 327**	**%**	**Bedroom of the child**	**N = 327**	**%**
temporary release (parental leave etc.)	14	4.4				no devices	181	55.4
			no	259	79.2	TV (I)	11	3.4
unemployed	11	3.5	yes	68	20.8	TV	49	15.0
apprentice	4	**1.3**	**Current use of medication**	**N = 327**	**%**	Games console (I)	34	10.4
**marital status**	**N = 327**	**%**	no	233	71.3	Games console	18	5.5
living with partner	273	83.5	yes	94	28.7	DVD-player	25	7.6
living alone	54	16.5	**Verbal communication (≥4 years)**	**N = 320**	**%**	Computer (I)	72	22.0
						Computer	8	2.4
			no	26	15.4	Tablet (I)	67	20.5
			yes	294	84.6	Tablet	8	2.4

*Note*. (I) with internet.

**Table 5 behavsci-09-00056-t005:** Codes and their frequency of occurrence.

	TV/DVD % (n)	PC/Video Games % (n)	Smartphone % (n)	Internet % (n)	Total % (n)
Wellbeing	64.02 (226)	75.77 (172)	57.90 (99)	53.62 (111)	63.47 (608)
Source of information	9.07 (32)	3.08 (7)	13.45 (23)	32.85 (68)	13.57 (130)
Joint family activities	12.46 (44)	3.08 (7)	1.17 (2)	0.48 (1)	5.64 (54)
Safety	3.12 (11)	4.85 (11)	11.70 (10)	3.38 (7)	5.12 (49)
Social communication/exchange	0.28 (1)	3.08 (7)	10.53 (18)	2.90 (6)	3.34 (32)
Babysitter	5.64 (20)	1.32 (3)	0.59 (1)	0.48 (1)	2.61 (25)
Reinforcement	1.13 (4)	4.85 (11)	1.17 (2)	1.93 (2)	2.19 (21)
Purpose in life	1.13 (4)	1.76 (4)	0.59 (1)	0.97 (2)	1.15 (11)
Social Acceptance/be part of a group	0.28 (1)	0.44 (1)	2.34 (4)	1.45 (3)	0.94 (9)
Background noise	1.70 (6)	0.44 (1)	0.59 (1)	-	0.84 (8)
Social learning tool	0.85 (3)	0.88 (2)	-	0.48 (1)	0.63 (6)
Advancement of other functional areas	0.28 (1)	0.44 (1)	-	1.45 (3)	0.52 (5)
	100.0 (353)	100.0 (227)	100.0 (171)	100.0 (207)	100.0 (985)

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
