# Peer review of "Caregiver Reports of Screen Time Use of Children with Autism Spectrum Disorder: A Qualitative Study"

_behavsci, 2019, doi:10.3390/bs9050056_

Round 1
Reviewer 1 Report
This is an interesting study on why Children and Youths with Autism Spectrum Disorder use Screen Media in their Leisure Time, and there is much to praise. The length of the interviews is to be commended. Nice clear description of analysis. The discussion is overall excellent, sound and comprehensive. The amount of data the authors have managed to include also makes this a very interesting read. Good that you considered age related differences. Functionality may also relate to whether or not the child had siblings, which you did not consider. The English is very good and does not need to be edited for publication but there are small indications that it is written by a non-native speaker.
However, while I think the study should be published, I do have major concerns that the information is indirect. While the authors do comment on this briefly in the discussion and the title refers to parents, it would be better to explain early on why the children themselves were not asked (even if they have learning disability some interesting information could be gained). I disagree with the mitigation on this in the discussion. The autism community is very active in its promotion of inclusion and participation in research, certainly in the UK (see Fletcher-Watson, S., Adams, J., Brook, K., Charman, T., Crane, L., Cusack, J., … Pellicano, E. (2019). Making the future together: Shaping autism research through meaningful participation. Autism, 23(4), 943–953.). Reporting on behalf of someone else (second hand information) is always less helpful than getting the information directly but this is especially the case with children with autism who may not have communicated their reasons for using media to their parents. Not all parents will have a deep understanding of autism, indeed it could be argued that only if you have autism can you truly understand it, but even without that view, there will be potential unreliability from parents’ reports and there will not be a clear pattern to the bias.
Background
L 38. It would be useful to have the 3 dimensions explained to understand how they should be considered, and not just functionality. Authors also need to explain how these link and consider in their discussion the effect of this on their findings, given that they only considered one construct.
The section on functionality 1.2 should be better organised so it is clear which are ASD and non-ASD studies and to contrast them better, even despite the aims of the paper. It is frustrating to hear that some studies say media use is good and some that it is bad, without any critique of the studies or expansion of the differences. I suspect that children with ASD also use video games because of their familiarity and predictability, indeed there is some research to support that. A distinction between learning disabled children with ASD and non-learning disabled should be made as the former may play them as a repetitive device (and your data support this as per lines 219-220). The type of media that can improve communication and social skills should be better differentiated (again despite the study aims).
Findings
The 2ndquote for relaxation is, to me, something other than relaxation, e.g. rather to reduce anxiety, indeed relaxation in both quotes is more about wellbeing. I wonder if the meaning behind the German word Gemütlich could have been the original conceptualisation as it fits better. Similarly, the ‘purpose in life’ quote fits this idea better. This does affect a lot of your paper so you need to provide quotes that justify ‘relaxation’ better or change it all to wellbeing which would be my preference on the basis of what I have seen in the paper – or have an extra code of wellbeing if there are clearer examples of relaxation. Your discussion about the relaxation code also aligns better with wellbeing/reducing anxiety.
Some smaller comments:
Instead of ‘[20] said’ , name the author then add the citation after as [20]
L 64 and also for positive uses (non-group)
L69 typo This, also specify what university and give ethics number.
Given that this study was undertaken in Germany, which has a policy approach to autism that differs from some other countries, and especially given where recruitment was from, there should be a paragraph on the context of autism in Germany, which would be expected to affect functionality.
Table 3 title – some exemplary (typo)
I presume common agreement l127 is 0.6 kappa?
You need to say whether the two coders coded everything and then compared and if so, to correct discrepant items, was one coder more influential than the other?
It is interesting that agreement of what seem quite concrete concepts was so low at first and some discussion of this would be useful.
L280 – some children with ASD are obsessive about fact finding and not just for homework, which should be mentioned.
Good discussion of different ASD profiles.
While I have suggested no English edits, there are a few errors. They are not significant and are more related to a Germanic style rather than making the text hard to understand.
Author Response
Please find the point-by-point response attached.

Reviewer 2 Report
Article Review
Why do Children and Youths with Autism Spectrum Disorder use Screen Media in their Leisure Time?: A Qualitative Study on Parental Perspectives
Consider a title change: Caregiver reports of screen time use of children with Autism Spectrum Disorder
This is an insightful study that adds to our knowledge base in ASD through the lens of a qualitative study.
Introduction: Change ‘school performance’ to ‘academic performance’
Change ‘lifes’ to ‘lives’
Methods line 1: Correct the grammar in methods and subsequent sections. Please review for correct formatting.
An elaboration of exact protocol for statistical methods is warranted.
Considerations: This is a relatively small sample size of children with ASD. It may be more appropriately termed a pilot investigation into the role of screen time in ASD. There is no control group. Where do you draw the line between problematic vs. unproblematic behaviors as related to screen time? You did not conduct in-home observations, but relied on parent report which has its own limitations. Please do not separate out ASD into subcomponents based on the older DSM IV. For example, the term Autism Spectrum Disorder now encompasses Asperger’s syndrome according to the DSM V.
Author Response

(The authors gave the same response as above.)
